# Multi-level Adaptive Contrastive Learning for Knowledge Internalization in Dialogue Generation

**Chenxu Yang**[1,2], **Zheng Lin**[1,2]*, **Lanrui Wang**[1,2], **Chong Tian**[4], **Liang Pang**[3],
**Jiangnan Li**[1,2], **Qirong Ho**[4], **Yanan Cao**[1,2], **Weiping Wang**[1,2]

[1]Institute of Information Engineering, Chinese Academy of Sciences, Beijing, China
[2]School of Cyber Security, University of Chinese Academy of Sciences, Beijing, China
[3]Institute of Computing Technology, CAS    [4]Mohamed bin Zayed University of AI
{yangchenxu,wanglanrui,lijiangnan,linzheng,wangweiping}@iie.ac.cn
{refrainkon,hoqirong}@gmail.com, pangliang@ict.ac.com

## Abstract

Knowledge-grounded dialogue generation aims to mitigate the issue of text degeneration by incorporating external knowledge to supplement the context. However, the model often fails to internalize this information into responses in a human-like manner. Instead, it simply inserts snippets of the provided knowledge into generic responses. As a result, the generated responses tend to be tedious, incoherent, and in lack of interactivity which means the degeneration problem is still unsolved. In this work, we find that such copying-style degeneration is primarily due to the weak likelihood objective, which allows the model to "cheat" the objective by merely duplicating knowledge snippets in a superficial pattern matching manner based on overlap. To overcome this challenge, we propose a Multi-level Adaptive Contrastive Learning (MACL) framework that dynamically samples negative examples and subsequently penalizes degeneration behaviors at both the token-level and sequence-level. Extensive experiments on the WoW dataset demonstrate the effectiveness of our approach across various pre-trained models and decoding strategies. [1]

## 1 Introduction

In recent years, pre-trained language models using transformer architectures have made remarkable strides in open-domain generation tasks (Lewis et al., 2020; Raffel et al., 2020; Zhang et al., 2020; Roller et al., 2020). However, these models still struggle with dull and repetitive outputs, a problem commonly termed as neural text degeneration (Holtzman et al., 2020; Welleck et al., 2019).

To address text degeneration in dialogue, Dinan et al. (2018) proposed to equip interlocutors with external knowledge as additional support to enrich the informativeness of responses, which is known

---

*Zheng Lin is the corresponding author.
[1]The code is available at https://github.com/iie-ycx/MACL.

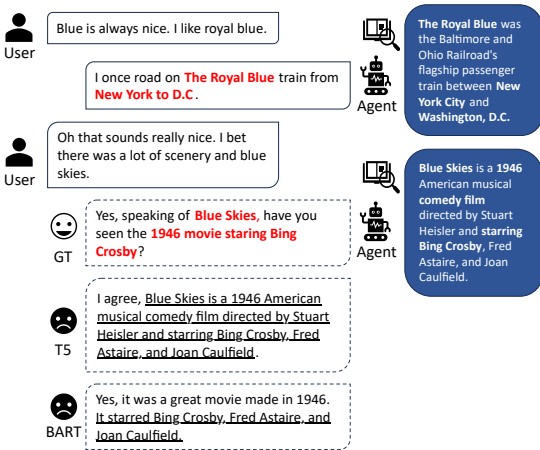

Figure 1: An example shows how current models make rigid use of knowledge and cause the responses incoherent with context. Both T5-large and BART-large are finetuned on WoW dataset with MLE.

as the Knowledge-Grounded Dialogue Generation (KGDG). Recent methods have put much emphasis on the knowledge selection subtask, aiming to provide models with the most suitable knowledge (Kim et al., 2020; Chen et al., 2020; Zheng et al., 2020; Yang et al., 2022).

However, simply introducing golden knowledge as a ground source does not indeed mitigate the problem of degeneration. We observe that existing models often duplicate knowledge snippets to construct responses to meet the informativeness requirement, which can lead to unnatural and contextually incoherent responses. For example, the generated contents in Figure 1 simply duplicate the provided knowledge about the "blue skies" movie, resulting in a tedious response incoherent with user's utterance. Consequently, we need to consider not only the introduction of knowledge, but also how to effectively integrate it into responses. **We term the superficially knowledge duplicating "Knowledge Regurgitation"** as it mirrors how the model absorbs an entire knowledge sentence

and regurgitates it in the response without genuine comprehension. **This issue can be seen as a task-specific copying-style manifestation of text degeneration.** We test some pre-trained models with different parameter scales, and discover that the Knowledge Regurgitation is common among them. To quantify its severity, we design two automated metrics: PoD and KuD. Based on a series of experiments, we posit that degeneration is caused by the ineffectual design of the training objective, which allows models to "cheat" the MLE objective by merely duplicating knowledge snippets to construct responses. The model, therefore, improperly converges to superficial pattern matching based on overlap, given that knowledge sentences share spurious correlations (token overlaps) with ground-truth responses and exhibit high semantic fluency.

In this paper, to tackle the aforementioned issues, we propose a novel approach known as Multi-level Adaptive Contrastive Learning. This method effectively mitigates Knowledge Regurgitation through contrastive training at both the token and sequence levels. At the token level, we enhance the negative training paradigm (Welleck et al., 2019) by employing a dynamic negative token sampling method and reweighting the unlikelihood loss according to model sensitivity. In this way, our approach penalizes tokens that continuously appear in the knowledge but are not the targets to cut off the potential shortcuts. At the sequence level, we first employ a group beam search strategy (Vijayakumar et al., 2018) to sample negative responses from the degenerator's predictions, then use a novel metric as an oracle function to score them. Finally, samples demonstrating obvious degeneration are selected as hard negatives for the InfoNCE loss to distance them from their prefix in the representation space. It can expose the model to potential degeneration mistakes that happen at the inference stage and help the model learn to avoid them.

Our contributions are summarized as follows:

- We explore a unique degeneration phenomenon in KGDG, termed "Knowledge Regurgitation", which is confirmed by a series of preliminary experiments in popular pre-trained language models.

- We propose a Multi-level Adaptive Contrastive Learning (MACL) framework, which is designed to effectively internalize knowledge at both token and sequence levels.

- We conduct extensive experiments and provide a detailed analysis to validate the effectiveness of our method, showing a substantial improvement on both automatic and human evaluation metrics.

## 2 Preliminaries

In this section, we provide a brief introduction to the KGDG task and the Unlikelihood Training (UT) loss, which serves as the foundation for our token-level contrastive learning loss.

### 2.1 Task Formulation

The knowledge-grounded dialogue generation task encompasses two relatively independent sub-tasks: knowledge selection and knowledge-aware response generation. Knowledge selection aims at selecting the most appropriate piece of knowledge, denoted $\hat{\mathbf{k}}$, from a given knowledge pool $\mathbf{KP} : \{\mathbf{k_1}, \mathbf{k_2}, \ldots, \mathbf{k_{|KP|}}\}$. Assuming that knowledge $\hat{\mathbf{k}}$ has been selected by an efficient knowledge selector, we mainly focus on its utilization. Given the dialogue context $\mathbf{u} = \{u_1, u_2, \ldots, u_m\}$ and the related knowledge $\hat{\mathbf{k}} = \{k_1, k_2, \ldots, k_s\}$, the goal is to generate an engaging and informative knowledge-infused response $\mathbf{y} = \{y_1, y_2, \ldots, y_{|\mathbf{y}|}\}$.

Given a KGDG dataset (Dinan et al., 2018) $\mathcal{D} = \{(\mathbf{u}^{(i)}, \mathbf{k}^{(i)}, \mathbf{y}^{(i)})\}$ derived from a collection of multi-turn human interactions, the standard method for training a sequence-to-sequence model involves concatenating the context with the knowledge as the model's input sequence and applying maximum likelihood estimation (MLE) to minimize:

$$\mathcal{L}_{\text{MLE}}(\mathbf{u}, \mathbf{k}, \mathbf{y}) = -\sum_{t=1}^{|\mathbf{y}|} \log p(y_t | \mathbf{u}, \mathbf{k}, y_{<t}),$$
$$(1)$$

where $\mathbf{u}$ is the context (golden dialogue history and user utterance at the current turn) , $\mathbf{k}$ is the golden knowledge, and $y_t$ is the $t$-th token of $\mathbf{y}$.

### 2.2 Unlikelihood Training

To address the problem of neural text degeneration, Welleck et al. (2019) proposed the unlikelihood training method, combining a token-level unlikelihood objective with MLE. The core idea involves selecting a set of negative token candidates, denoted as $\mathcal{C}_t$, at each training step and reducing their prediction probability $p(y_c)$, while concurrently increasing the probability of ground-truth

tokens $p(y_t)$. Negative candidates typically consist of tokens that have already been generated, alleviating the degeneration phenomenon where generated texts contain undesirable repetitions at various levels and high frequency tokens appear excessively.

$$\mathcal{L}_{\mathrm{UL}}(\mathcal{C}, \mathbf{x}, \mathbf{y}) = -\sum_{t=1}^{|\mathbf{y}|} \sum_{y_c \in \mathcal{C}_t} \log(1 - p(y_c|\mathbf{x}, y_{<t})).$$
$$\mathcal{C}_t = \{\mathbf{x}, y_1, y_2, \ldots, y_{t-1}\}/\{y_t\}, \qquad (2)$$

where $\mathcal{C}_t$ is a subset of the vocabulary and $\mathbf{x}$ is the prefix sequence.

The MLE loss aims to model the groud-truth sequence probability distribution, while the unlikelihood loss corrects undesired patterns. The overall objective in unlikelihood training is mixed by them as follows:

$$\mathcal{L}_{\mathrm{UT}} = \mathcal{L}_{\mathrm{MLE}} + \alpha \mathcal{L}_{\mathrm{UL}}, \qquad (3)$$

where $\alpha$ is a hyper-parameter varying from different tasks and datasets.

## 3  Knowledge Regurgitation

Although current pre-trained language models have demonstrated robust performance in generating fluent dialogue responses, they fail to align with human-like patterns of knowledge integration.

We conduct a series of preliminary experiments to demonstrate the presence of text degeneration problems in some pre-trained language models of various sizes. Specifically, we quantify the severity of this issue by comparing human and model performance based on our newly proposed metrics. The results are presented in Table 1 and Figure 2, which demonstrates the model's severe knowledge regurgitation and singular knowledge utilization patterns.

Dup-n and Proportion of Longest Common Subsequence tokens (PLCS) are mainly used to measure knowledge snippets duplication. Dup-n represents the proportion of samples with n-grams co-occurring in knowledge and response and is computed as follows:

$$\text{Dup-}\mathbf{n} = \frac{1}{|D|} \sum_{(\mathbf{k}, \mathbf{y}) \in D} \mathbb{1}(\mathbf{n}\text{-gram}(\mathbf{k}) \cap \mathbf{n}\text{-gram}(\mathbf{y})), \quad (4)$$

where $D$ denotes the dataset, $\mathbf{k}$ denotes the golden knowledge, and $\mathbf{y}$ denotes the generated response.

|  | Dup-16 | Dup-32 | PLCS | mKP-1 |
|---|---|---|---|---|
| T5-base | 72.82% | 61.84% | 73.68% | 82.25% |
| T5-large | 66.33% | 57.55% | 69.88% | 80.04% |
| BART-base | 70.55% | 61.76% | 73.20% | 81.67% |
| BART-large | 54.93% | 46.34% | 63.05% | 75.91% |
| GPT2-large | 45.52% | 40.53% | 53.85% | 71.10% |
| GPT2-xl | 54.45% | 48.18% | 60.07% | 74.76% |
| LLaMa-7B | 34.21% | 37.34% | 44.09% | 60.56% |
| human | 6.53% | 2.37% | 24.47% | 47.83% |

Table 1: Part of the preliminary experiment results on the WoW dataset.

For PLCS, it is calculated as ratio of the length of longest common sub-sequence (LCS) of response and knowledge to the length of response as follows:

$$\text{PLCS} = \frac{1}{|D|} \sum_{(\mathbf{k}, \mathbf{y}) \in D} \frac{|\text{LCS}(\mathbf{k}, \mathbf{y})|}{|\mathbf{y}|}. \qquad (5)$$

The aforementioned metrics approximately measure the frequency of a degenerated knowledge utilization pattern where only snippets of the provided knowledge are inserted into a generic response without substantial integration.

To highlight the gap in knowledge utilization patterns between humans and models, we further employed metrics related to the precision of knowledge grams (mKP-n), which calculates the mean ratio of knowledge tokens in response to all the response tokens. It is performed as follows:

$$\text{mKP-}\mathbf{n} = \frac{1}{|\mathrm{D}|} \sum_{(\mathbf{k}, \mathbf{y}) \in \mathrm{D}} \text{KP-}\mathbf{n}. \qquad (6)$$

$$\text{KP-}\mathbf{n} = \frac{|\mathbf{n}\text{-gram}(\mathbf{k}) \cap \mathbf{n}\text{-gram}(\mathbf{y})|}{|\mathbf{n}\text{-gram}(\mathbf{y})|}. \qquad (7)$$

A comparison of the performance of each MLE-finetuned PLM with human dialogue, presented in Table 1, makes it evident that the duplication frequency of models surpasses that of humans. Although humans occasionally replicate professional knowledge and famous quotes to construct knowledgeable responses, such usage is less prevalent in casual chitchat, as shown in the last row of Table 1. Models, however, tend to misinterpret the duplication pattern superficially and apply it across various contexts.

In chit-chat scenarios, it is important to strike a balance with the number of knowledge tokens used in responses. Excessive knowledge tokens can reduce the interactivity of the response, potentially

diminishing users' desire to continue the conversation. From the data presented in Table 1, we can observe that knowledge tokens account for approximately 50% of human responses, whereas the proportion in model-generated responses is too high. Additionally, the knowledge precision distributions between humans and models are illustrated in Figure 2. It highlights that humans exhibit a more diverse usage of given knowledge across various contexts, as evidenced by their uniform knowledge precision distribution. In contrast, the distributions of pre-trained language models exhibit a distinct pattern, with the probability mass predominantly concentrated on the higher percentage side.

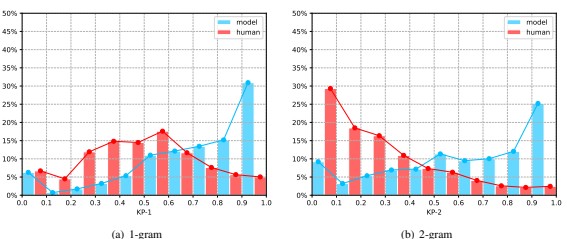

(a) 1-gram        (b) 2-gram

Figure 2: Knowledge Precision Distribution of human responses (red) and machine-generated responses (blue) on the WoW dataset.

## 4 Our Approach

We propose a novel multi-level contrastive learning method that dynamically samples negative examples and subsequently penalizes degeneration behaviors at both the token and sequence levels.

**Token-level Contrastive Learning**    The motivation behind the token-level loss is to break the generation inertia more effectively (refer to the Figure 3 for a detailed explanation). Our improvements over basic unlikelihood training method are mainly twofold: dynamic negative sampling and dynamic unlikelihood loss reweighting. The detailed loss function is presented in the following formula:

$$\mathcal{L}_{\text{token}}^t(\mathcal{C}_t, \mathbf{u}, \mathbf{k}, \mathbf{y}) = -\log p(y_t|\mathbf{u}, \mathbf{k}, y_{<t})$$
$$-\alpha \sum_{y_c \in \mathcal{C}_t} \beta(y_c) \log(1 - p(y_c|\mathbf{u}, \mathbf{k}, y_{<t})), \quad (8)$$

where $\mathcal{C}_t$ and $\beta(y_t)$ are calculated as follows:

$$\mathcal{C}_t = \arg\max(p_w(\mathbf{k})).$$
$$\beta(y_c) = \cos((1 - p_c)\pi) + 1, \quad (9)$$

where $p_c$ is predictive probability of $y_c$, and it is a scalar detached from the computational graph.

We empirically investigate the selection strategy of negative examples in our study. Given the objective of the knowledge-grounded dialogue task is to effectively integrate knowledge into the response, and that these knowledge tokens are essential potential candidates, punishing all knowledge without distinction is not appropriate. Our goal is to eliminate shortcuts leading to knowledge duplication while keep the knowledge integrating ability. To achieve this, we propose to dynamically select negative tokens based on the model's sensitivity, thereby reducing the probability of tokens where the model is more likely to err. Specifically, we adopt a strategy of selecting the knowledge token with the highest prediction probability as the negative token. We also explored other strategies, such as sampling by probability, random selection, but the adopted strategy outperforms the others. We conjecture that this is due to our strategy targeting the prediction chains of the knowledge snippet (shortcuts), effectively disrupting it.

As for the dynamic reweighting strategy, we considered two reasons. Firstly, Jiang et al. (2020) highlight the importance of applying differentiable weights to individual token losses by proposing Token Focal Loss inspired by Lin et al. (2018)'s work. In light of this, we enhance the unlikelihood loss by introducing an additional control parameter $\beta(y_c)$ to dynamically reweight the punishment strength for different tokens. By doing so, we suppress the gradients of easy tokens while amplifying the gradients of hard tokens, leading to faster and improved convergence during training.

Secondly, Lin et al. (2021) expressed concern that the vanilla negative training method might cause the model to decrease the probability of the target token $p_*$ to reduce the gradient norm $|\nabla \mathcal{L}_a|$ (10) during the final stages of training, particularly when $\alpha$ is excessively large (refer to the Appendix B for a detailed explanation). Our approach mitigates the problem of that inverse optimization and allows the loss function to converge to a lower value as the introduction of $\beta(y_c)$ facilitates the gradual decrease of the weight of $\mathcal{L}_{\text{UL}}$ (2).

Ground-Truth Token $(i = i^*)$ :
$$\nabla \mathcal{L}_a = \frac{\partial \mathcal{L}}{\partial p_i}\frac{\partial p_i}{\partial a_i} = 1 - p_i(1 - \alpha\beta(y_c)\frac{p_c}{1 - p_c}). \tag{10}$$

**Sequence-level Contrastive Learning**    Regarding the sequence-level contrastive loss, An et al. (2023) point out that contrastive learning provides

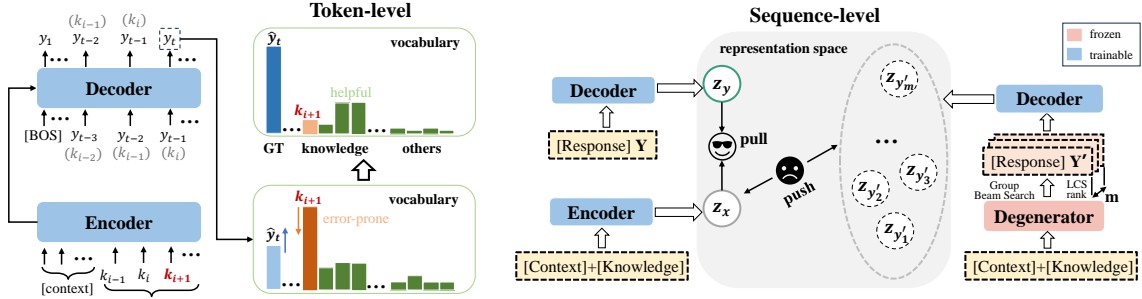

Figure 3: An overview of the MACL framework. The mark in the top left corner of the figure indicates that certain response tokens are identical to the knowledge tokens enclosed in brackets. Owing to the generation inertia, that is, the shortcut, the model predicts an exceptionally high probability for the next knowledge token, which surpasses the probability of the ground-truth token. MACL breaks the knowledge snippet chain effectively and outputs more reasonable distributions. $z_x$ represents the source sequence (context and knowledge), and $z_y$ stands for its target sequence (response). The feature representations are derived from pooling the outputs of the encoder (source sequence) or decoder (target sequence), which are both differentiable in this context.

a novel solution to alleviate the exposure bias problem. Arora et al. (2022) suggest that the degeneration problem is a result of exposure bias, which motivates us to address this issue by leveraging the sequence-level contrastive learning method during the training phase. By exposing the model to negative targets exhibiting degeneration, we aim to help the model learn to avoid predicting them.

$$\mathcal{L}_{\text{seq}} = \\ -\log \frac{e^{\cos(\mathbf{z}_x, \mathbf{z}_y)}}{\sum_{y' \in B} e^{\cos(\mathbf{z}_x, \mathbf{z}_{y'})} + \sum_{y^- \in H} \mu e^{\cos(\mathbf{z}_x, \mathbf{z}_{y^-})}}, \quad (11)$$

where $B$ are from-batch negative samples, $H$ are generated hard negative samples and $\mu$ is used to reweight the hard negatives.

We begin by collecting a diverse set of utterances that exhibit significant degeneration problems using the following negative sampling method. Firstly, we train the PLM with MLE to obtain a degenerator, which generates responses exhibiting degeneration phenomena. We then employ a group beam search strategy (beam size $b$) to acquire negative responses from this degenerator. Finally, these responses are evaluated using a metric that measures the degree of duplication, serving as an oracle function for scoring. We utilize the length of the Longest Common Sub-sequence (LCS) with knowledge sequence as the oracle function. We retain the top $m$ degenerated response samples.

Following the collection procedure, we drive the distance between the source sequence representation and the negative target sequence representation in a contrastive way. The $\mathcal{L}_{\text{seq}}$ loss provides

the model with a negative supervision signal on the shortcut paths that generated these sequences. Both hard negative samples and from-batch negative samples are retained, but we assign higher weights to the former to emphasize their importance.

The final objective function is defined as:

$$\mathcal{L}_{final}(\theta) = \mathcal{L}_{\text{token}}(\theta) + \lambda \mathcal{L}_{\text{seq}}(\theta). \quad (12)$$

See the Appendix E for the pseudo-code of the entire training procedure.

## 5 Experiments

### 5.1 Experimental Setup

**Dataset.** Three common datasets have been typically used to evaluate the KGDG task: Holl-E (Moghe et al., 2018), CMU_DoG (Zhou et al., 2018), and WoW (Dinan et al., 2018). However, the topics in the first two datasets are limited to movies, with much of the knowledge being composed of movie reviews in the form of dialogues. This is not in line with our goal of exploring the internalization of retrieved world knowledge. The CMU_DoG dataset did not label golden knowledge, so there is no way to tell if the knowledge introduced is correct. Through observations and experiments, we find that the Holl-E dataset is also not suitable for conducting the evaluation of knowledge regurgitation, and we put the detailed reasons and experimental results in the Appendix F. Consequently, we choose the WoW dataset for our experiments. See the Appendix A for a brief introduction to the chosen WoW dataset.

**Baseline Methods.** We compare our MACL framework with vanilla MLE and several state-of-the-art (SOTA) methods that address the issue of text degeneration.

**NT:** Welleck et al. (2019) proposed the concept of unlikelihood loss, combining it with MLE loss.

**Scalegrad:** Lin et al. (2021) modified the gradient of the MLE, encouraging the model to use novel tokens. we designate knowledge tokens as non-novel tokens.

**ND:** Li et al. (2022) developed a novel training paradigm known as negative distillation, designed to steer the model away from undesirable degenerated responses. We utilize the MLE-finetuned model as the negative teacher.

**CTloss:** Jiang et al. (2022) put forward a new contrastive token learning objective. This objective aims to promote label tokens in the ranking at each step while demoting negative tokens, leaving other irrelevant tokens unaffected.

**SimCTG:** Su et al. (2022) introduced a contrastive objective designed to learn discriminative and isotropic token representations by increasing the distances between distinct tokens' representations.

**Implementation Details.** We use PyTorch (Paszke et al., 2019) framework to implement our work. For the implementation of PLMs BART, T5 and GPT-2, we utilize the open-source Hugging Face transformers (Wolf et al., 2020). The whole model is optimized with Adam (Kingma and Ba, 2014) algorithm. We set the learning rate to 1e-5 and training batch size to 16, train up to 15 epochs, and select the best checkpoints based on performance on the validation set. Some hyper-parameters are set as follows: $\alpha = 4, \lambda = 1, \mu = 2, b = 32, m = 16$. At the inference stage, we utilize a decent knowledge selector designed by Yang et al. (2022) to select knowledge first and utilize it to generate response to fit the KGDG. The decoding strategy is set to beam search with a beam size of 3.

**Evaluation Metrics.** We choose perplexity (PPL) of the ground-truth responses, BOW Embedding (Avg., Ext.) (Liu et al., 2016), BLEU-1(Papineni et al., 2002), Knowledge Utilization Difference (KUD), and Porportion of Degenerated samples (PoD) as automatic metrics for evaluation. The latter two are the metrics we propose to measure knowledge regurgitation and the difference from humans in knowledge utilization pattern.

Given that the primary characteristic of degeneration is the duplication of knowledge fragments,

we propose considering samples with a Proportion of Longest Common Sub-sequence (PLCS) greater than 70% as degenerated samples. We manually annotate a subset of samples from the test set and calculate the precision of the automated metric, PoD. The annotators are given the following instructions: A generated response will be classified as degenerated if it 1) evidently replicates external knowledge and 2) produces content that is visibly unnatural and contextually incoherent. The results in Table 4 demonstrate effectiveness of the PoD metric. To compare the gap between different methods and humans in knowledge utilization patterns, we introduce the metric KUD, which is defined as:

$$\text{KUD} = \text{MAE}(\text{P}_{\text{h}}(\text{KP-}\mathbf{1})||\text{P}_{\text{g}}(\text{KP-}\mathbf{1})), \quad (13)$$

where $\text{P}_{\text{h}}$ is the distribution of human response, and $\text{P}_{\text{g}}$ is the distribution of generated response.

For human evaluation, We randomly select 50 responses in test seen set and 50 responses in test unseen set. We conducted an aspect-based pairwise preference test. Specifically, for a given context, we paired our model's response with a response from the baselines and asked five well-educated annotators to choose the superior response based on the following four aspects: (1) **Coherence**: which model generates more contextually coherent responses; (2) **Engagingness**: which model generates more interesting responses; (3) **Informativeness**: which response contains more knowledge; (4) **Interactiveness**: which model generates more interactive responses that make the user want to continue the conversation. We compute Fleiss' kappa value (Fleiss, 1971) among different annotators to measure their agreement.

## 5.2 Experimental Results

Table 2 presents the automatic evaluation results on the WoW dataset. Our method, MACL, significantly mitigates knowledge regurgitation, reducing the proportion of degeneration by 13.42% on the test seen set and 16% on the test unseen set compared to the strongest baseline, Scalegrad. In terms of KUD metric, the enhancements of MACL are impressive, exhibiting a stark reduction in discrepancies from humans. It is about ten times better than the best performing baseline NT on the test seen dataset and five times superior on the test unseen dataset. The distribution of 1-gram and 2-gram knowledge, as shown in Figure 4, closely aligns with that of humans, indicating that MACL

| | Test Seen | | | | | | Test Unseen | | | | | |
|---|---|---|---|---|---|---|---|---|---|---|---|---|
| | PPL | PoD(%) | KUD | Avg. | Ext. | BLEU-1 | PPL | PoD(%) | KUD | Avg. | Ext. | BLEU-1 |
| MLE | 23.784 | 52.31 | 7.28 | 0.842 | 0.428 | 23.7 | 26.146 | 54.78 | 7.62 | 0.839 | 0.422 | **22.6** |
| NT | 22.858 | 25.16 | 4.85 | 0.848 | 0.436 | 23.3 | 25.008 | 26.97 | 5.89 | 0.845 | 0.430 | 22.2 |
| ND | 53.653 | 29.68 | 5.34 | **0.852** | 0.428 | 22.0 | 63.907 | 25.40 | 5.28 | 0.847 | 0.419 | 20.4 |
| CTloss | 27.599 | 37.56 | 5.29 | 0.839 | 0.431 | 23.3 | 32.704 | 35.92 | 5.12 | 0.838 | 0.425 | 22.2 |
| SimCTG | 24.466 | 52.56 | 7.25 | 0.842 | 0.428 | **23.8** | 28.679 | 54.07 | 7.65 | 0.838 | 0.422 | **22.6** |
| Scalegrad | 25.204 | 21.35 | 5.02 | 0.843 | 0.436 | 22.3 | 29.461 | 23.82 | 5.88 | 0.840 | 0.428 | 21.7 |
| MACL | **20.897*** | **7.93*** | **0.52*** | **0.852** | **0.438** | 22.5 | **23.973*** | **7.82*** | **1.11*** | **0.848** | **0.435*** | 22.0 |

Table 2: Automatic Evaluation results on the WoW dataset (BART-large). The best results are highlighted with **bold**. "*" denotes that the improvement to the best baseline is statistically significant (t-test with $p$-value $< 0.01$).

| Comparisons | Coherence | | | | Engagingness | | | | Informativeness | | | | Interactiveness | | | |
|---|---|---|---|---|---|---|---|---|---|---|---|---|---|---|---|---|
| | Win | Lose | Tie | $\kappa$ | Win | Lose | Tie | $\kappa$ | Win | Lose | Tie | $\kappa$ | Win | Lose | Tie | $\kappa$ |
| MACL vs. NT | 22.8 | 6.8 | 70.4 | 0.596 | 29.2 | 12.2 | 58.6 | 0.535 | 20.8 | 12.4 | 66.8 | 0.472 | 39.4 | 10.8 | 49.8 | 0.561 |
| MACL vs. SimCTG | 35.2 | 5.0 | 59.8 | 0.586 | 42.2 | 6.6 | 51.2 | 0.522 | 22.0 | 25.0 | 53.0 | 0.445 | 45.2 | 8.4 | 46.4 | 0.647 |
| MACL vs. Scalegrad | 22.0 | 4.0 | 74.0 | 0.481 | 29.2 | 7.6 | 63.2 | 0.618 | 23.4 | 9.0 | 67.6 | 0.474 | 32.2 | 6.8 | 61.0 | 0.481 |

Table 3: Human Evaluation results on the WoW dataset (%). The result is statistically significant with $p$-value $< 0.05$.

| Method | PoD(aotu) | PoD(human) | PoD Precision |
|---|---|---|---|
| MLE | 52% | 55% | 94.2% |
| MACL | 8% | 9% | 87.5% |

Table 4: The proportion of samples with degeneration phenomenon annotated by humans on the 100 samples extracted, and the precision of our proposed metric PoD.

successfully internalizes knowledge into responses and achieves a similar ability to utilize knowledge as humans.

While the baseline methods effectively address traditional text degeneration, their impact on alleviating knowledge regurgitation is not as significant. This suggests that knowledge regurgitation is distinct from conventional repetition-style degeneration. We believe this is because KGDG ensures that knowledge is integrated into the response, and simply reducing the prediction probability of all tokens in the prefix is insufficient. This highlights the effectiveness of MACL's dynamic sampling and dynamic penalty mechanisms in adapting to the KGDG task.

In terms of conventional metrics that evaluate content quality, MACL achieves state-of-the-art (SOTA) performance in terms of perplexity, average (Avg.), and extrema(Ext.). This indicates that the quality of the generated responses is high and comparable to human performance. However, for the BLEU metric, which is based on gram overlap, MACL performs slightly worse than the baselines. We attribute this to the fact that the longer responses, which reveal knowledge regur-

gitation, contain a higher number of knowledge tokens. As a result, the hit rate of knowledge tokens in the response is higher, leading to inflated scores for degenerated responses, despite their contextual incoherence. In the example provided in Table 10, we observed that the BLEU-1 score of the MLE-generated responses is higher than that of the MACL-generated responses (30.77 > 17.65).

More experimental outcomes based on other pretrained models and decoding strategies are detailed in the Appendix C.

The human-based evaluation results are shown in Table 3. Notably, MACL consistently outperforms all the compared methods. MACL effectively internalizes knowledge into its generated responses. As a result, the responses produced by MACL are more natural and human-like, avoiding direct narration. Regarding the Context Coherence metric, MACL maintains a relatively better focus on the user, carefully balancing the attention between the user's utterance and the knowledge during the generation stage. While MACL may lose some information compared to SimCTG's responses, it is important to note that excessive knowledge is inappropriate in chit-chat scenarios. The kappa results indicate a moderate level of agreement among the annotators.

### 5.3 Ablation Study

To analyze the sources of improvement achieved by MACL, we conducted an ablation study. The ablation results, shown in Table 5, indicate that all of these design components contribute to the

|            | Test Seen |        |      |       |       |        | Test Unseen |        |      |       |       |        |
|------------|-----------|--------|------|-------|-------|--------|-------------|--------|------|-------|-------|--------|
|            | ppl       | PoD(%) | KUD  | Avg.  | Ext.  | BLEU-1 | ppl         | PoD(%) | KUD  | Avg.  | Ext.  | BLEU-1 |
| MACL       | **20.897**| **7.93** | **0.52** | **0.852** | **0.438** | 22.5 | **23.973** | **7.82** | **1.11** | **0.848** | **0.435** | 22.0 |
| -Reweighting | 23.663  | 12.14  | 2.71 | 0.850 | 0.437 | 22.9   | 26.260      | 14.09  | 3.30 | 0.847 | 0.432 | 22.3   |
| -TCL       | 23.441    | 37.76  | 6.49 | 0.844 | 0.431 | **23.5** | 26.473    | 38.05  | 7.01 | 0.840 | 0.423 | **22.4** |
| NaiveSCL   | 21.592    | 22.37  | 1.57 | 0.849 | 0.436 | 23.0   | 24.136      | 21.19  | 1.92 | 0.846 | 0.431 | 22.3   |
| -SCL       | 21.702    | 22.45  | 1.55 | 0.848 | 0.436 | 23.1   | 24.424      | 20.92  | 1.93 | 0.844 | 0.429 | 22.2   |

Table 5: Ablation study on the WoW dataset. -Reweighting denotes removing the dynamic reweighting method compared to MACL. -TCL denotes removing the token-level contrastive learning compared to MACL. NaiveSCL denotes using only from-batch negative samples in InfoNCE compared to MACL. -SCL denotes removing the sequence-level contrastive learning method compared to MACL.

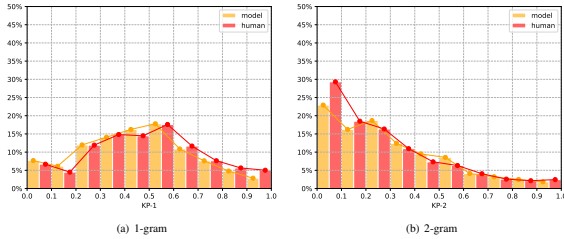

Figure 4: Knowledge Precision Distribution of human responses (red) and MACL-generated responses (orange) on the WoW dataset.

effectiveness of MACL. Removing the dynamic reweighting factor results in a significant increase in the perplexity metric, demonstrating that dynamic weighting is indeed beneficial for mitigating the inverse optimization problem. Token-level contrastive learning plays a crucial role in aligning the model's ability to utilize knowledge with that of humans, as removing this loss function leads to a significant degradation in the KUD metric. Both the token-level and sequence-level contrastive learning functions are key in suppressing knowledge regurgitation, removing either of them results in an increase in the PoD metric. The performance improvement brought by sequence-level contrastive learning mainly stems from the selection of hard negative examples. When only from-batch negatives are used, the negative examples are easily distinguishable from the ground truth, and the model cannot effectively learn additional capabilities through the loss.

### 5.4   Case Study

To better evaluate the performance of response generation, we selected examples generated by MLE, NT, Scalegrad, and MACL for comparison. In the example provided in Table 10, the baselines are unable to avoid the pattern of inserting snippets of the provided knowledge into generic responses,

resulting in contextually incoherent and unengaging responses. MACL, on the other hand, not only expresses its own opinion on the football player but also supplements relevant knowledge about him. The examples in Table 11 demonstrate that MACL's dynamic negative sampling mechanism reasonably selects negative candidates and penalizes them without disrupting complete quotes. See the Appendix D for more cases and analyses.

## 6   Related Work

**Knowledge-grounded dialogue generation** With the increased functional demands for open-domain dialogue robots, many source-grounded dialogue generation tasks appear on researchers' radar (Wang et al., 2022; Lim et al., 2023; Zhang et al., 2018), especially knowledge-grounded conversations. The hot spot of research is mainly concentrated on how to improve the performance of knowledge selection (Sun et al., 2023; Xu et al., 2022). Zhan et al. (2021) proposed a collaborative latent variable (CoLV) model to integrate the two subtasks simultaneously in collaborative latent spaces; Zhao et al. (2022) established a multi-reference KGDG dataset and devised a span-based variational model; Yang et al. (2022) introduced the topic shift information into knowledge selection subtask.

**Neural Text Degeneration** Neural text degeneration refers to the problem that the generated texts from the language model tend to be dull and incoherent, contain undesirable repetitions at different levels (Holtzman et al., 2020; Li et al., 2020). The existing methods mainly alleviates this problem from two aspects: decoding strategy and training strategy (Su et al., 2022; Lagutin et al., 2021). Holtzman et al. (2020) found that the distribution of candidate tokens obtained by existing language models had unreliable long-tails, so they cut off them to reduce the occurrence of incoherent se-

quences; Li et al. (2020) adjusted the negative training method to alleviate some generation problems in open domain dialogue tasks like inconsistency and contradiction. Su et al. (2022) attributed degeneration to the anisotropy of token representation in vector space, and designed a solution from both the training and decoding stage.

# 7    Conclusion

In this paper, we investigate a distinctive degeneration phenomenon in Knowledge-Grounded Dialogue Generation referred to as Knowledge Regurgitation, which is prevalent in pre-trained language models. To address this challenge, we present a novel solution called Multi-level Adaptive Contrastive Learning (MACL). Our approach tackles the problem by dynamically sampling negative examples and penalizing degeneration behaviors at both the token-level and sequence-level. Experiments on the WoW dataset demonstrate that our approach significantly mitigates knowledge regurgitation.

## Limitations

MACL effectively addresses the issue of Knowledge Regurgitation. However, we acknowledge certain limitations in our work:

(1) Due to limited computational resources, we have focused on demonstrating the effectiveness of our method on pre-trained language models with Encoder-Decoder structures that have less than 1 billion parameters. However, our method can indeed address the degeneration problem in lightweight chit-chat models, and we plan to explore degeneration in larger language models (LLMs) in future work.

(2) As existing datasets did not align with the specific scenario we wanted to explore, we solely evaluated our method on the WoW dataset. Although the dataset size may not be large enough, it provided valuable insights for our research.

(3) Our sequence-level contrastive loss involves generating negative examples during the training stage, which requires multiple calls to the pre-trained language model to calculate the hidden states of negative targets. We have not yet optimized our algorithm code to run in parallel, resulting in a decrease in training speed.

## Ethics Statement

The benchmark dataset we used in our experiments, WoW (Dinan et al., 2018), is a well-regarded, open-source dataset collected by crowdsourced workers. It was compiled with rigorous adherence to user privacy protection protocols, ensuring the exclusion of any personal information. Furthermore, our proposed approach consciously upholds ethical standards and societal fairness, ensuring that no prejudice is introduced. For our human evaluation component, all participants were volunteers who were provided with comprehensive information about the research's purpose, ensuring informed consent. In addition, all participants received fair and reasonable compensation for their contributions.

## Acknowledgments

This work was supported by the National Natural Science Foundation of China (No. 61976207), the National Natural Science Foundation of China (NSFC) under Grants No. 62276248, and the National Social Science Foundation of China (No. 21AZD145).

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

# A  Details of Dataset

The WoW data was sourced from a crowdsourcing website. During data collection, the user side plays the role of the apprentice, and the agent side plays the role of the wizard. The wizard has access to the knowledge retrieved from Wikipedia as ground-source to generate informative responses, while the apprentice prefers speaking common utterances. The WoW dataset consists of 22,311 dialogues with 201,999 turns, which are divided into a training set, a validation set, a test seen set, and a test unseen set. The topics in the test unseen set are ones that never appeared in the training set.

# B  Problem of Negative Training (NT)

As the gradient-based optimization progresses, it is expected that the gradient of the loss approaches zero around a minimum. Therefore, the probability of the ground-truth token $p_i$ should increase towards 1 to decrease the gradient norm $|\nabla \mathcal{L}_a|$ and achieve convergence to a value close to 0 during the training stage.

However, in equation (14), when $p_i > \frac{1}{1+\alpha}$, the value of the gradient norm $|\nabla \mathcal{L}_a|$ becomes larger than 1. Consequently, the training procedure reduces $p_i$ to minimize the gradient norm, which contradicts the optimization principle. This issue prevents the loss function from converging during the later stages of training.

Ground-Truth Token $(i = i^*)$ :
$$\nabla \mathcal{L}_a = \frac{\partial \mathcal{L}}{\partial p_i} \frac{\partial p_i}{\partial a_i} = 1 - p_i(1 - \alpha \frac{p_c}{1 - p_c}) \quad (14)$$

## C   More Experimental Results

As observed in Table 8, MACL works on the T5-large as well, effectively reducing the knowledge regurgitation. Furthermore, in addition to the beam search decoding strategy with a beam size of 3, we also experimented with beam search using a beam size of 5, greedy decoding, and Nucleus Sampling. The results in Table 9 show that larger beam sizes are more prone to degeneration. Although nucleus sampling could alleviate knowledge regurgitation, it is far less effective than MACL. It also illustrates that our approach is compatible with these decoding strategies as it mitigates the degeneration further.

## D   More Cases

In the example provided in Table 12, MACL effectively incorporates the knowledge about the movie "Blue Skies" into its response. It starts by acknowledging that there are indeed beautiful views and blue skies along the route of the Royal Blue train, and then proceeds to mention relevant content about the movie with the same name. On the other hand, the baselines fail to internalize the knowledge and generate responses that are incoherent and lack interactivity.

## E   Details of the Training Algorithm

See the pseudo-code below for details.

## F   More results on the Holl-E dataset

The WoW dataset is well collected, with a specific focus on engagingness and interactiveness. The collectors crafted responses by integrating grounded knowledge naturally, and they were forbidden to duplicate knowledge snippets for saving time. Compared to that, the collection of Holl-E dataset is relatively rough. There is apparent replication between the ground-truth response and the introduced knowledge in it (which means the ground-truth response is not a ideal response with knowledge internalization), it is unsuitable for evaluating the prevention of knowledge regurgitation. It sometimes takes movie comments as both external knowledge (input) and ground-truth responses (target) during data collection, leading to problematic ground-truth responses. The comparison between the two datasets are shown in Table 6.

The experimental results on the PoD metric show that MACL still effectively mitigate knowledge

|  | PoD | Dup-16 | Dup-32 | PLCS | mKP-1 |
|---|---|---|---|---|---|
| WoW | 5.54% | 6.53% | 2.37% | 24.47% | 47.83% |
| Holl-E | 73.43% | 75.68% | 70.86% | 78.91% | 82.36% |

Table 6: Comparison between the WoW dataset and the Holl-E dataset.

|  | PPL | PoD(%) | KUD | Avg. | Ext. | BLEU-1 |
|---|---|---|---|---|---|---|
| MLE | 2.151 | 67.34% | 0.64 | 0.890 | **0.599** | 69.2 |
| NT | 2.214 | 61.77% | 0.89 | 0.887 | 0.593 | 70.4 |
| ND | 3.576 | 58.21% | 1.12 | 0.889 | 0.595 | 68.8 |
| CTloss | 2.653 | 66.34% | 0.70 | 0.887 | 0.594 | 69.1 |
| SimCTG | 2.218 | 68.62% | **0.61** | 0.888 | 0.595 | 69.5 |
| Scalegrad | 2.334 | 59.98% | 0.96 | **0.891** | 0.598 | 70.7 |
| MACL | **2.117** | **49.63%** | 1.50 | 0.889 | 0.595 | **71.3** |

Table 7: Automatic Evaluation results on the Holl-E dataset (BART-large). The best results are highlighted with **bold**.

replication phenomenon (-17.71%). However, the dataset remains unsuitable for evaluating the knowledge regurgitation problem in terms of the results of the other metrics. All the baseline methods achieved notably low perplexity and remarkably high BLEU-1 score. It indicates the excessive similarity between knowledge input and generation targets. Besides, the generated response with less severe knowledge replication results in a broader discrepancy in knowledge utilization capacity (higher KUD value). The phenomenon illustrates problematic ground-truth responses.

**Algorithm 1** MACL Training algorithm: Given a knowledge-grounded dialogue dataset $\langle \mathcal{X}, \mathcal{Y} \rangle$, a pre-trained language model $\mathcal{M}$; return a finetuned model $\mathcal{M}^*$.

1: **procedure** TRAINDEGENERATOR($\mathcal{M}$)
2:    Update the parameters of $\mathcal{M}$ with $\nabla\theta\mathcal{L}_{MLE}$ until convergence and Get the degenerator $\mathcal{M}_{de}$
3:    **return** $\mathcal{M}_{de}$
4: **procedure** NEGATIVESAMPLING($\mathcal{M}, \langle \mathbf{x}, \mathbf{y} \rangle, m$)
5:    $\mathbf{y}^{(1:b)} \leftarrow$ Do BeamSearch to get $b$ samples      ▷ group beam search algorithm
6:    $\mathbf{y}^{(1:m)} \leftarrow$ Do oracle measurement $o(\mathbf{y}^{(1:b)}, \mathbf{x}_k)$ for each element and get the top $m$ samples
7:    **return** $\mathbf{y}^{(1:m)}$
8: **procedure** TRAIN($\mathcal{M}, \langle \mathcal{X}, \mathcal{Y} \rangle$)
9:    $\theta \leftarrow$ Parameters of $\mathcal{M}$, $b \leftarrow$ beam size
10:    $\mathcal{M}_{de} =$ TrainDegenerator($\mathcal{M}$)      ▷ the degenerator is frozen since now
11:    **while** not convergence **do**
12:       $\langle \mathbf{x}^{(1:k)}, \mathbf{y}^{(1:k)} \rangle \leftarrow$ A minibatch of $k$ datapoints from $\langle \mathcal{X}, \mathcal{Y} \rangle$
13:       $\mathcal{L}_{token} \leftarrow$ Get token-level contrastive loss for $\langle \mathbf{x}^{(1:k)}, \mathbf{y}^{(1:k)} \rangle$
14:       $\mathbf{y}^{(1:k;1:m)} =$ NegativeSampling $(\mathcal{M}_{de}, \langle \mathbf{x}^{(1:k)}, \mathbf{y}^{(1:k)} \rangle, b)$
15:       $\mathbf{y}^{(1:k;1:k+m)} \leftarrow$ Append $m$ degeneration samples to $\mathbf{y}^{(1:k)}$
16:       $\mathcal{L}_{seq} \leftarrow$ Get sequence-level contrastive loss for $\langle \mathbf{x}^{(1:k)}, \mathbf{y}^{(1:k;1:k+m)} \rangle$
17:       update parameters using $\nabla\theta(\mathcal{L}_{\text{token}} + \lambda\mathcal{L}_{\text{seq}})$
18:    **return** $\mathcal{M}^*$

| | Test Seen | | | | | | Test Unseen | | | | | |
|---|---|---|---|---|---|---|---|---|---|---|---|---|
| | ppl | PoD(%) | KUD | Avg. | Ext. | BLEU-1 | ppl | PoD(%) | KUD | Avg. | Ext. | BLEU-1 |
| MLE | 15.018 | 69.54 | 8.05 | 0.843 | 0.426 | 23.1 | 16.773 | 71.00 | 8.19 | 0.836 | 0.419 | 21.9 |
| NT | 14.890 | 36.94 | 5.95 | 0.851 | 0.434 | 22.7 | 16.569 | 38.38 | 6.13 | 0.844 | 0.428 | 21.7 |
| ND | 33.653 | 32.58 | 5.34 | 0.842 | 0.428 | 21.0 | 47.931 | 34.55 | 5.28 | 0.841 | 0.419 | 20.4 |
| CTloss | 17.063 | 37.56 | 5.29 | 0.839 | 0.431 | 23.3 | 18.788 | 35.92 | 5.12 | 0.838 | 0.425 | 22.2 |
| SimCTG | 15.420 | 66.68 | 7.71 | 0.843 | 0.426 | 23.2 | 17.251 | 68.6 | 8.05 | 0.836 | 0.419 | 22.1 |
| Scalegrad | 15.089 | 36.97 | 4.72 | 0.841 | 0.427 | 22.4 | 16.863 | 36.62 | 8.28 | 0.834 | 0.420 | 21.8 |
| MACL | **13.534** | **10.93** | **0.95** | **0.855** | **0.439** | 22.5 | **14.565** | **11.82** | **1.36** | **0.848** | **0.435** | 22.0 |

Table 8: Automatic Evaluation results on the WoW dataset (T5-large).

| decoding strategy | method | Test Seen | | | | | | Test Unseen | | | | | |
|---|---|---|---|---|---|---|---|---|---|---|---|---|---|
| | | ppl | PoD(%) | KUD | Avg. | Ext. | BLEU-1 | ppl | PoD(%) | KUD | Avg. | Ext. | BLEU-1 |
| Beam | MLE | 23.784 | 62.86 | 7.40 | 0.839 | 0.425 | 22.9 | 26.146 | 64.11 | 7.76 | 0.835 | 0.418 | 21.7 |
| Search | MACL | 20.897 | 11.74 | 1.22 | 0.848 | 0.437 | 22.5 | 23.973 | 13.01 | 1.76 | 0.846 | 0.434 | 22.0 |
| Greedy | MLE | 23.784 | 43.03 | 5.92 | 0.850 | 0.437 | 23.8 | 26.146 | 45.21 | 6.12 | 0.844 | 0.430 | 23.4 |
| Search | MACL | 20.897 | 3.32 | 1.35 | 0.851 | 0.437 | 21.5 | 23.973 | 4.41 | 1.71 | 0.841 | 0.433 | 20.8 |
| Nucleus | MLE | 23.784 | 20.03 | 3.02 | 0.842 | 0.427 | 22.3 | 26.146 | 19.74 | 3.08 | 0.845 | 0.420 | 21.6 |
| Sampling | MACL | 20.897 | 1.54 | 0.75 | 0.847 | 0.423 | 21.3 | 23.973 | 2.37 | 0.59 | 0.846 | 0.425 | 20.9 |

Table 9: Automatic Evaluation results on the WoW dataset across various decoding strategies (BART-large). The beam size of the beam search is set to $5$ and the pre-chosen threshold $p$ for nucleus sampling is set to $0.9$.

| Context | Thierry Henry is one of my all time favorite players. What about you? |
|---|---|
| Knowledge | Thierry Daniel Henry (born 17 August 1977) is a retired French professional footballer who played as a forward and is the second assistant manager of the Belgium national team. |
| GT | He was good. He is a retired French professional footballer. |
| MLE | Yes, he is a retired French professional footballer who played as a forward and now he is the second assistant manager of the Belgium national team. |
| NT | Yes, he is a great player. He is now the second assistant manager of the Belgium national team. |
| Scalegrad | Yes, he is a retired French professional footballer. |
| MACL | I love him too! He was a great forward born on August 17th, 1977! |

Table 10: An example for case study.

| Context | (1) I am retired now. Are you familiar with the work of Mehr and Meyer, well known psychologists? (2) I am not. Could you tell me about them? |
|---|---|
| Knowledge | According to Maehr and Meyer, "Motivation is a word that is part of the popular culture as few other psychological concepts are." |
| GT | They said that "Motivation is a word that is part of the popular culture as few other psychological concepts are." Do you agree with what they said? |
| MLE | Well, according to Maehr and Meyer, "Motivation is a word that is part of the popular culture as few other psychological concepts are." |
| NT | Well according to Maehr and Meyer, "Motivation is a word that is part of the popular culture as few other psychological concepts are." |
| Scalegrad | Maehr and Meyer are psychologists. They argue that motivation is a word that is part of the popular culture as few other psychological concepts are. |
| MACL | They say that "Motivation is a word that is part of the popular culture as few other psychological concepts are." |

Table 11: An example for case study.

| Context | (1) Blue is always nice. I like royal blue. (2) I once road on The Royal Blue train from New York to D.C. (3) Oh that sounds really nice. I bet there was a lot of scenery and blue skies. |
|---|---|
| Knowledge | Blue Skies is a 1946 American musical comedy film directed by Stuart Heisler and starring Bing Crosby, Fred Astaire, and Joan Caulfield. |
| GT | Yes, speaking of Blue Skies, have you seen the 1946 movie staring Bing Crosby? |
| MLE | It was a 1946 American musical comedy film directed by Stuart Heisler and starring Bing Crosby, Fred Astaire, and Joan Caulfield. |
| NT | I bet there were lots of blue skies in that movie with Bing Crosby and Fred Astaire. |
| Scalegrad | Yes, there was a lot of scenery and blue skies in the movie Blue Skies which was released in 1946. |
| MACL | Yes it was! And Blue Skies was also a comedy film directed by Stuart Heisler! |

Table 12: An example for case study.