# OpenReview forum: "Multi-level Adaptive Contrastive Learning for Knowledge Internalization in Dialogue Generation"
_EMNLP/2023/Conference — EMNLP 2023 Main_

### Official Review · Reviewer_pQyx · 2023-07-21

**Soundness:** 4

**Excitement:**

3: Ambivalent: It has merits (e.g., it reports state-of-the-art results, the idea is nice), but there are key weaknesses (e.g., it describes incremental work), and it can significantly benefit from another round of revision. However, I won't object to accepting it if my co-reviewers champion it.

**Paper Topic And Main Contributions:**

The paper applies text degeneration techniques to knowledge-grounded dialogue generation. Specifically, they tackle the problem whereby models learn to simply regurgitate knowledge wholesale, without appropriately integrating it into a natural human response. They approach this via a Multi-level Adaptive Contrastive Learning framework, which penalises degeneration both at the token and sequence level. They show SoTA results across both human and automated evaluation.

**Questions For The Authors:**

(a) Do the same knowledge regurgitation problems occur if the model is trained on predicted knowledge from a knowledge selection module, rather than the golden knowledge?

(b) Can you motivate why no knowledge-grounded dialogue baselines are included (or clarify which baselines count as knowledge-grounded dialogue baselines)?

Post-rebuttal edit:
In light of the significant additional work conducted by the authors in response to my questions/concerns, I have moved my soundness rating to a 4. To clarify my point around end-to-end methods being less likely to regurgitate knowledge, I had in mind the architectures that treat the knowledge as a latent variable to be marginalised over [1] [2]--here, the generator is trained on the retriever's distribution, rather than the golden knowledge.

1) Shuster, Kurt et al. “Retrieval Augmentation Reduces Hallucination in Conversation.” Conference on Empirical Methods in Natural Language Processing (2021).

2) Zhang, Yizhe et al. “RetGen: A Joint Framework for Retrieval and Grounded Text Generation Modeling.” AAAI Conference on Artificial Intelligence (2021).

**Reasons To Accept:**

(i) The paper identifies an interesting problem--namely, models learning to trivially copy knowledge into the response. This strategy can lead to models performing well in most automated metrics (e.g. PPL, bleu etc.) which can make it hard to identify with current metrics. The authors present a range of diversity metrics which show significant differences between the level of copying used by humans versus models.

(ii) The authors support their claims across a range of automated and human metrics and ablations.

(iii) The application of text degeneration techniques to knowledge-grounded dialogue is quite novel, and both the token-level and sequence-level contrastive learning each seem to provide a sizeable improvement to performance according to the ablation.

**Reasons To Reject:**

(i) Section 4 could benefit from additional clarity, especially the part on token-level contrastive learning. It isn't clear whether the β(y_c) in equation 8 is the same β as in line 285 or equation 10. It also isn't clear how TFL fits into the loss function, as the only equation it appears in is equation 9. Also, InfoNCE loss is never fully explained--I assume it refers to Lseq, but then I'm not sure of the motivation behind having a separate term for it.

(ii) The paper treats knowledge selection (KS) and knowledge-aware response generation (RG) as relatively independent (line 124). I think however that RG is highly depdendent on KS, given many SoTA architecture train both models jointly end-to-end. In that case, the RG module might be less likely to just regurgitate knowledge, as it is knows the KS module is not 100% accurate. So far as I can tell, all of the experiments involve only training RG on the golden knowledge, in which case it is less surprising that the model learns to regurgitate. I think it would be a stronger paper therefore if you could show that RG had the same problem when trained on the distribution of KS, rather than the golden distribution.

(iii) The baselines are all chosen from text degeneration literature, rather than knowledge-grounded dialogue. I think this is an issue because the relationship between the context and response is very different from the relationship between the knowledge and the response. There is typically little term-level overlap between context and response, so penalising terms in the context can work well. However, there is significant overlap between knowledge and response, as is the premise of the paper, so it is not clear at all that these techniques will perform well in that setting. From the results it is not clear that the baselines surpass even the basic MLE method, other than in diversity metrics.

**Reproducibility:**

4: Could mostly reproduce the results, but there may be some variation because of sample variance or minor variations in their interpretation of the protocol or method.

**Reviewer Confidence:**

4: Quite sure. I tried to check the important points carefully. It's unlikely, though conceivable, that I missed something that should affect my ratings.

---

> ### Author Rebuttal · Authors · 2023-08-28
>
> Thank you very much for your careful review! We are glad that you believe our approach "is quite novel", that you thought the problem we explored was "an interesting problem", etc. We hope our response could alleviate your concerns and your valuable advice will be fully
> incorporated in the final version.
>
> **Q1: Additional experiment on training with predicted knowledge**
>
> **A1:** Thank you for your insightful questions. Your suggestion of employing predictive knowledge for training the response generator is very interesting. We conducted the experiment as you proposed.  But here we must correct your misconception about end-to-end training. Indeed, "many SoTA architecture train both models jointly end-to-end". This does not indicate that the RG module was trained on predicted knowledge if you check their source codes. **Basically, the RG module was also trained on golden knowledge in the end-to-end scenerio**[1][2][3][4][5].
>
> | Bart-large      | PPL    | PoD    | KUD  | Avg.  | Ext.  | BLEU-1 |
> | --------------- | ------ | ------ | ---- | ----- | ----- | ------ |
> | WoW test seen   | 24.871 | 27.61% | 5.13 | 0.842 | 0.428 | 22.3   |
> | WoW test unseen | 27.208 | 29.34% | 5.45 | 0.838 | 0.423 | 21.4   |
>
> Although knowledge regurgitation issue was somewhat mitigated, it didn't work out so well. Besides, **other metrics (including BLEU, perplexity, Avg., and Ext.) exhibited a corresponding decline**. By comparison, the MACL approach proved more effectual in mitigating knowledge regurgitation, without abandoning the model's capacity for knowledge integration. We do not think training with predicted knowledge is a solution for knowledge regurgitation for two reasons.
>
> **1)** By introducing some noises, the model avoids excessive reliance on external knowledge. However, this approach comes with drawbacks too: **the model's capacity to integrate knowledge is compromised, resulting in a notable reduction in the generated responses' knowledgeability**.
>
> **2)** As you rightly pointed out, the accuracy of the knowledge selector is far from 100%. However, it may approach 100% when better methods are proposed, and the issue of knowledge regurgitation will persist. Thus, we maintain that our work retains its intrinsic value.
>
> **We believe that the intrinsic reason for knowledge regurgitation is the ineffective design of the MLE training objective**. The model continued to converge towards superficial pattern matching based on overlap, as the selected knowledge still shares spurious correlations with ground-truth responses.
>
> Nevertheless, we sincerely appreciate your question(ii) as lacking corresponding experiments and explanations, our paper is not solid enough.
>
> | WoW Test Seen(Bart-large) | PPL    | PoD    | KUD  | Avg.  | Ext.  | BLEU-1 |
> | ------------------------- | ------ | ------ | ---- | ----- | ----- | ------ |
> | TAKE-Bart                 | 24.871 | 27.61% | 5.13 | 0.842 | 0.428 | 22.3   |
>
>
> | WoW Test Unseen(Bart-large) | PPL    | PoD    | KUD  | Avg.  | Ext.  | BLEU-1 |
> | --------------------------- | ------ | ------ | ---- | ----- | ----- | ------ |
> | TAKE-Bart                   | 27.208 | 29.34% | 5.45 | 0.838 | 0.423 | 21.4   |
>
>
>
> **Q2: Clarifying our choice of Baselines**
>
> **A2:** **Although the baselines are all chosen from text degeneration, we concatenated user's query with golden knowledge in the input before response generation. We believe this treatment is consistent with the KGDG generation process.** In this setting, we denoted the MLE approach as the KGDG baselines because the optimization for response generators of the KGDG baselines relies on MLE loss functions.
>
> **Q3: Clarity improvement**
>
> **A3:** Thanks for your valuable suggestions aimed at enhancing the clarity of Section 4. The β in these places refers to the same parameter. While Eq(9) has been introduced to underscore the inspiration drawn from focal loss in our method, it does introduce a degree of redundancy. Therefore, we plan to remove Eq(9) in subsequent versions. Additionally, $L_{seq}$ is derived through the improvement of InfoNCE loss, a widely employed contrastive loss. We are committed to modifying them in the upcoming version of the manuscript.
>
> [1] Initiative-Aware Self-Supervised Learning for Knowledge-Grounded Conversations. 2021.
>
> [2] Sequential latent knowledge selection for knowledge-grounded dialogue. 2020.
>
> [3] Bridging the gap between prior and posterior knowledge selection for knowledge-grounded dialogue generation. 2020
>
> [4] DukeNet: A Dual Knowledge Interaction Network for Knowledge-Grounded Conversation. 2020
>
> [5] TAKE: Topic-shift Aware Knowledge sElection for Dialogue Generation. 2022

---

### Official Review · Reviewer_XHDH · 2023-08-01

**Soundness:** 3

**Excitement:**

4: Strong: This paper deepens the understanding of some phenomenon or lowers the barriers to an existing research direction.

**Paper Topic And Main Contributions:**

This paper focuses on knowledge-grounded dialogue generation scenarios.

It first examines the "Knowledge Regurgitation" phenomena that exist in many LMs, where the generation results usually have the same substrings as the given knowledge. Such a copy mechanism makes responses more dull and more unnatural.

To tackle this issue, authors propose Multi-level Adaptive Contrastive Learning (MACL). It introduces negative samples that contain noticeable overlaps with the knowledge texts, then applies both token-level and sentence-level contrastive learning along with dynamic loss weights to fine-tune the pre-trained LM.

Experiments are conducted on the WoW dataset, where MACL achieves similar n-gram accuracy as the baselines, while a significant improvement on the degeneration of knowledge text pieces .

**Questions For The Authors:**

A. Why do you mention Eq.(9)? Seems that it is not used in the MACL method. According to my understanding, it is for different weighting on the loss of different tokens. But such an idea is very straightforward which somehow makes this part redundant. Besides, what does p_t mean in Eq.(9)?

B. How do you obtain z_x and z_y and make them differentiable in the whole procedure? Are they the representations of some special tokens in pre-trained LMs?

C. In L254-257, do C^t and C_t represent the same thing? "where p_c is prediction probability of C_t”. So what is p_c? According to my understanding, C_t is a set and p_c is a scalar. Maybe p_c is the probability of y_t?

D. In Table 4, how do human annotators make these judgments? Could you provide more details like the instruction and definition?

**Reasons To Accept:**

1. The paper is well-motivated. Directly copying text pieces from the original knowledge, rather than making genuine comprehension and proper paraphrasing, is a common shortcut learned by LMs which degenerates the generated responses. This paper not only mentions this issue but also verifies its existence using experiments on multiple general pre-trained LMs, along with pre-defined metrics.

2. The proposed method, MACL, is also reasonable as it is inspired by the observed phenomena. The additional token-level unlikelihood loss can prevent it from directly copying text pieces from the original knowledge context, while the dynamic weights can pose heavier weights on unexpected tokens with higher possibilities. On the other hand, sentence-level contrastive learning loss can further make the generated responses more dissimilar to the bad cases.

**Reasons To Reject:**

1. The experiments can be further extended to make the paper more solid. 1) Only one dataset is included, which means its extensibility and generalization remain unverified. Maybe authors can add the experimental results on the rest two datasets mentioned in Section 5.1. 2) Only relatively smaller pre-trained LMs are involved. Since LLMs are hot topics in recent research, including LLMs and illustrating their performance can promote the value of this paper. 3) I think such a degeneration also appears in personalized dialogue generation, it would be better if related content can be included.

**Reproducibility:**

3: Could reproduce the results with some difficulty. The settings of parameters are underspecified or subjectively determined; the training/evaluation data are not widely available.

**Reviewer Confidence:**

3: Pretty sure, but there's a chance I missed something. Although I have a good feel for this area in general, I did not carefully check the paper's details, e.g., the math, experimental design, or novelty.

**Typos Grammar Style And Presentation Improvements:**

I did not carefully check the typos and may miss some of them.

L68: "mDup and KPDD-n" only appear here and never appear later, maybe typos.

L569: alleviates -> alleviate

Missing articles, like L257 "prediction" and L595 "degeneration" and so on.

---

> ### Author Rebuttal · Authors · 2023-08-28
>
> Thank you very much for the useful and constructive feedback! We are glad that you liked many aspects of our paper (thinking that it is well-motivated, that our method MACL is reasonable, etc.). We have fixed the typos and grammatical errors you mentioned in the updated version of the paper following your valuable advice. We hope the following responses will adequately address your concerns.
>
> **Q1: Clarifying our choice of Dataset**
>
> **A1:**
> Thank you for your constructive comment. We have implemented experiments on the Holl-e dataset to validate the extensibility and generalization of the problem we addressed. We will include them in the appendix of a subsequent version of the paper to showcase the extensibility and generalization of the knowledge regurgitation issue as you suggested.
>
> | Holl-e (Bart-large) | PPL   | PoD    | KUD  | Avg.  | Ext.  | BLEU-1 |
> | ------------------- | ----- | ------ | ---- | ----- | ----- | ------ |
> | MLE                 | 2.151 | 67.34% | 0.64 | 0.890 | 0.599 | 69.2   |
> | NT                  | 2.214 | 61.77% | 0.89 | 0.887 | 0.593 | 70.4   |
> | ND                  | 3.576 | 58.21% | 1.12 | 0.889 | 0.595 | 68.8   |
> | CTloss              | 2.653 | 66.34% | 0.70 | 0.887 | 0.594 | 69.1   |
> | SimCTG              | 2.218 | 68.62% | 0.61 | 0.888 | 0.595 | 69.5   |
> | Scalegrad           | 2.334 | 59.98% | 0.96 | 0.891 | 0.598 | 70.7   |
> | MACL                | 2.117 | 49.63% | 1.50 | 0.889 | 0.595 | 71.3   |
>
> **The experimental results on the PoD metric show that MACL still effectively mitigate knowledge replication phenomenon (-17.71%)**. However, the dataset remains unsuitable for evaluating the knowledge regurgitation problem in terms of the results of the other metrics. All the baseline methods achieved notably low perplexity and remarkably high BLEU-1 score. It indicates the excessive similarity between knowledge input and generation targets. Besides, the generated response with less severe knowledge replication results in a broader discrepancy in knowledge utilization capacity (higher KUD value). The phenomenon illustrates problematic ground-truth responses.
>
> Now we explain our choices in detail. Basically, there are three widely used KGDG datasets: WoW[1], CMU_DoG[2], and Holl-e[3]. The reason we only present the WoW dataset results is not only this dataset "meet our requirements". Instead, we chose this dataset due to its superior quality and knowledge diversity. The WoW dataset is well collected, with a specific focus on engagingness and interactiveness. The collectors crafted responses by integrating grounded knowledge naturally, and they were forbidden to duplicate knowledge snippets for saving time.
>
> We did not adopt the Holl-E and CMU_DoG datasets for evaluation because **there is apparent replication between the ground-truth response and the introduced knowledge in Holl-E and CMU_DoG datasets (which means the ground-truth response is not a ideal response with knowledge internalization), they are unsuitable for evaluating the prevention of knowledge regurgitation**. The two datasets take movie comments as both external knowledge (input) and ground-truth responses (target) during data collection, leading to problematic ground-truth responses.
>
> Regarding CMU_DoG, it is even worse. The dataset did not label golden knowledge, so there is no way to tell if the knowledge introduced is correct. We investigate how to prevent knowledge regurgitation under the assumption that knowledge has been correctly selected. Therefore, we were unable to complete the experiment on this dataset.
>
> Due to page limits, we did not elaborate on the above reasons supporting our dataset selection within the paper. We plan to supplement them in the forthcoming edition.
>
> **Q2: Partial LLM results**
>
> **A2:** We only carried out fine-tuning of the llama[4] model with lora[5] subject to rebuttal time limits. **The experimental results indicate that LLMs suffer from the knowledge regurgitation issue similarly, with a noticeable gap persists between the generated response and human-annotated responses** (Almost **5 times worse** on the Dup-16 metric and **15 times worse** on the Dup-32 metric). In future endeavors, we intend to undertake more extensive experiments across various open-source LLMs.
>
> | WoW Test Seen | PoD    | Dup-16 | Dup-32 | PLCS  | KUD  | mKP-1 |
> | ------------- | ------ | ------ | ------ | ----- | ---- | ----- |
> | llama-MLE     | 26.45% | 34.21  | 37.34  | 44.09 | 3.97 | 60.56 |
> | human     | - | 6.53  | 2.37  | 24.47 | - | 47.83 |
>
> **Q3: personalized dialogue generation**
>
> **A3:** Regarding personalized dialogue generation, we concur with your observation that replication issues could also be present. We intend to carry out experiments using the **Persona-Chat** dataset and incorporate their results in the forthcoming version of the paper.
>
> **Q4: Responses for "Questions For The Authors"**
>
> **A4:**
>
> **A**. Eq(9) is introduced to illustrate that our approach draws inspiration from focal loss. As you rightly noted, this might lead to some redundancy in processing. Therefore, we plan to remove Eq(9) in subsequent versions. We appreciate your suggestion. Here, p_t signifies the predicted probability of the next target token.
>
> **B**. The caption of Figure 3 explicitly states that z_x and z_y are derived through mean pooling from sentence representations, and they retain their differentiability in this context.
>
> **C**. Indeed, C^t and C_t refer to the same entity. Additionally, p_c denotes the probability of y_t.
>
> **D**. Instructions: A generated response will be classified as degenerated if it **1) evidently replicates external knowledge and 2) produces content that is visibly unnatural and contextually incoherent**.
>
> Given that automated metrics can only assess condition 1), we augmented the evaluation with human judgement. The results indicate the merit of the automated PoD metric.
>
> We sincerely appreciate your opinion on Typos Grammar Style And Presentation Improvements.
>
>
> [1] ICLR2019 Wizard of Wikipedia: Knowledge-powered conversational agents.
>
> [2] EMNLP2018 A dataset for document grounded conversations.
>
> [3] EMNLP2018 Towards exploiting background knowledge for building conversation systems.
>
> [4] LLaMA: Open and Efficient Foundation Language Models. 2023.
>
> [5] LoRA: Low-Rank Adaptation of Large Language Models. 2022.

---

### Official Review · Reviewer_Q1NT · 2023-08-04

**Soundness:** 3

**Excitement:**

3: Ambivalent: It has merits (e.g., it reports state-of-the-art results, the idea is nice), but there are key weaknesses (e.g., it describes incremental work), and it can significantly benefit from another round of revision. However, I won't object to accepting it if my co-reviewers champion it.

**Paper Topic And Main Contributions:**

The paper proposes a Multi-level Adaptive Contrastive Learning (MACL) to address the problem of knowledge regurgitation in knowledge-grounded dialogue generation. The authors first identify the issue of copying-style degeneration in current models, where the model simply duplicates knowledge segments without genuine comprehension. They then propose MACL, which includes both token-level and sequence-level contrastive learning to penalize degeneration behaviors and improve knowledge utilization in the generated responses. The proposed approach is evaluated on the WoW dataset and compared against several baselines, showing significant improvements in terms of knowledge regurgitation, knowledge utilization, and content quality.

**Questions For The Authors:**

1) Can you provide more details on the implementation of the MACL framework, including the specific architecture, hyperparameters, and training process?

2) How knowledge selection is handled in experiments with MACL and other baseline models.

3) Experience has found that large language models do not suffer from such severe replication problems; is the knowledge echo problem mentioned by the authors only a problem with the WoW dataset?

**Reasons To Accept:**

1) The paper introduces an important problem in knowledge-grounded dialogue generation, which is the issue of knowledge regurgitation, and proposes a approach to mitigate this problem.

2) The proposed MACL framework includes both token-level and sequence-level contrastive learning, providing a comprehensive solution to address degeneration behaviors at different levels.

3) The paper provides a thorough evaluation of the proposed approach, comparing it against several baselines and conducting both automatic and human evaluations. The results show significant improvements in various metrics, demonstrating the effectiveness of MACL.

**Reasons To Reject:**

1) The paper lacks implementation details, which may hinder the reproducibility of the proposed approach. More information on the specific architecture, hyperparameters, and training process would be helpful.

2) The proposed method was validated on only one dataset (WoW) and the authors also mention in the data and limitations that only one dataset meets their requirements, does it prove that this is not a generic problem but a flaw in the WOW dataset？

3) The authors did not focus on knowledge selection and did not mention if golden knowledge was used for training. This may be unfair to compare with other baseline models.

**Reproducibility:**

4: Could mostly reproduce the results, but there may be some variation because of sample variance or minor variations in their interpretation of the protocol or method.

**Reviewer Confidence:**

4: Quite sure. I tried to check the important points carefully. It's unlikely, though conceivable, that I missed something that should affect my ratings.

---

> ### Author Rebuttal · Authors · 2023-08-28
>
> Thank you very much for your detailed review! We are glad that you believe our approach is "a comprehensive solution", that you thought the problem we explored is "an important problem", etc. Your thoughtful and valuable suggestions will be incorporated in the future version of our paper. The questions and problems will be properly addressed as we reply.
>
> **Q1: Clarifying our choice of Dataset**
>
> **A1:** Thank you for your constructive comment. We have implemented experiments on the Holl-e dataset to validate the problem we addressed is a generic problem instead of a flaw in the WOW dataset. We will include them in the appendix of a subsequent version of the paper to showcase the extensibility and generalization of the knowledge regurgitation issue as you suggested.
>
> | Holl-e (Bart-large) | PPL   | PoD    | KUD  | Avg.  | Ext.  | BLEU-1 |
> | ------------------- | ----- | ------ | ---- | ----- | ----- | ------ |
> | MLE                 | 2.151 | 67.34% | 0.64 | 0.890 | 0.599 | 69.2   |
> | NT                  | 2.214 | 61.77% | 0.89 | 0.887 | 0.593 | 70.4   |
> | ND                  | 3.576 | 58.21% | 1.12 | 0.889 | 0.595 | 68.8   |
> | CTloss              | 2.653 | 66.34% | 0.70 | 0.887 | 0.594 | 69.1   |
> | SimCTG              | 2.218 | 68.62% | 0.61 | 0.888 | 0.595 | 69.5   |
> | Scalegrad           | 2.334 | 59.98% | 0.96 | 0.891 | 0.598 | 70.7   |
> | MACL                | 2.117 | 49.63% | 1.50 | 0.889 | 0.595 | 71.3   |
>
> The experimental results on the PoD metric show that MACL still effectively mitigate knowledge replication phenomenon (-17.71%). However, the dataset remains unsuitable for evaluating the knowledge regurgitation problem in terms of the results of the other metrics.
> All the baseline methods achieved notably low perplexity and remarkably high BLEU-1 score. It indicates the excessive similarity between knowledge input and generation targets. Besides, the generated response with less severe knowledge replication results in a broader discrepancy in knowledge utilization capacity (higher KUD value). **The phenomenon illustrates problematic ground-truth responses.**
>
> Now we explain our choices in detail. Basically, there are three widely used KGDG datasets: WoW[1], CMU_DoG[2], and Holl-e[3]. The reason we only present the WoW dataset results is not only this dataset "meet our requirements". **Instead, we chose this dataset due to its superior quality and knowledge diversity. The WoW dataset is well collected, with a specific focus on engagingness and interactiveness**. The collectors crafted responses by integrating grounded knowledge naturally, and they were forbidden to duplicate knowledge snippets for saving time.
>
> We did not adopt the Holl-E and CMU_DoG datasets  for evaluation because **there is apparent replication between the ground-truth response and the introduced knowledge in them (which means the ground-truth response is not a ideal response with knowledge internalization), they are unsuitable for evaluating the prevention of knowledge regurgitation.**
> The two datasets take movie comments as both external knowledge (input) and ground-truth responses (target) during data collection, leading to problematic ground-truth responses.
>
> Regarding CMU_DoG, it is even worse. The dataset did not label golden knowledge, so there is no way to tell if the knowledge introduced is correct. We investigate how to prevent knowledge regurgitation under the assumption that knowledge has been correctly selected. Therefore, we were unable to complete the experiment on this dataset.
>
> Due to page limits, we did not elaborate on the above reasons supporting our dataset selection within the paper. We plan to supplement them in the forthcoming edition.
>
> Some of the low-quality data:
>
> ```
> tr_3_0 -> tr_3_1
>
> Query: Which scene did you like the most in the movie?
> Knowledge: I liked the one which the old man has a heart attack during a KY jelly wrestling match..
> Response: I liked the one which the old man has a heart attack during a KY jelly wrestling match
>
> tr_4_0 -> tr_4_1
>
> Query: What was your favorite scene in this movie?
> Knowledge: I liked the one which Jack is trolley cart while the bus was speeding so he could diffuse the bomb. It was exciting..
> Response: I liked the one which Jack is trolley cart while the bus was speeding so he could diffuse the bomb. It was exciting
> ```
>
> The following table compares the quality of ground-truth responses in both the Holl-e and WoW datasets, evaluated on metrics concerning knowledge regurgitation.
>
> |        | PoD    | Dup-16 | Dup-32 | PLCS   | mKP-1  |
> | ------ | ------ | ------ | ------ | ------ | ------ |
> | WoW    | 5.54%  | 6.53%  | 2.37%  | 24.47% | 47.83% |
> | Holl-e | 73.43% | 75.68% | 70.86% | 78.91% | 82.36% |
>
> **Q2: Clarifying the Fairness of comparison**
>
> **A2:** Your next concern is about whether the knowledge is fairly provided. For all approaches, golden knowledge is used in the training stage and a SoTA knowledge selector is used in the inference stage to select the knowledge for generating response. **The knowledge selection and other settings are the same, so we believe our comparison is fair.**
>
> **Q3: Does LLM suffers from knowledge regurgitation issue as well?**
>
> **A3:** large language models also suffer from replication problems in the KGDG scenario. Regarding the open-source fine-tuned LLM, we carried out fine-tuning of the llama[4] model with lora[5]. In future endeavors, we intend to undertake more extensive experiments across various open-source LLMs.
>
> | WoW Test Seen | PoD    | Dup-16 | Dup-32 | PLCS  | KUD  | mKP-1 |
> | ------------- | ------ | ------ | ------ | ----- | ---- | ----- |
> | llama-MLE     | 26.45% | 34.21  | 37.34  | 44.09 | 3.97 | 60.56 |
> | human     | - | 6.53  | 2.37  | 24.47 | - | 47.83 |
>
> **The experimental results indicate that a noticeable gap still exists between the llama-generated responses and human-annotated responses.** (Almost **5 times worse** on the Dup-16 metric and **15 times worse** on the Dup-32 metric)
>
> In the case of closed-source LLMs such as ChatGPT, we conducted tests using zero-shot prompting with provided instructions. Though the replication issue is largely mitigated, its outputs are excessively lengthy, which doesn't align with casual chitchat scenarios. Consequently, we assert that the problem of knowledge regurgitation remains a challenge.
>
> **Q4: Some implementation details missing in the paper**
>
> **A4:** Regarding model architecture, we embrace the original BART and T5 (encoder-decoder architecture). Our primary emphasis lies in training loss design and negative sample collection, without introducing new modules.
>
> We have provided all the hyperparameter configurations in L383 of the paper and the Supplementary Materials (our codes). We are committed to releasing our code on GitHub upon publication. Please let me know what other hyperparameters you'd like to know about. We will supplement them then.
>
> ```
> learning rate: 1e-5, inference batch size: 32, training scheduler: linear_schedule_with_warmup, warmup_ratio: 0.2, max length of dialogue context: 64, max length of knowledge sentence: 64, max length of response during training stage: 64. Some generation configs: max_new_tokens: 60, min_length: 10.
> ```
>
> As for the training process, we have provided detailed steps in algorithm 1 in the Appendix. You can learn more about it there.
>
>
> [1] ICLR2019 Wizard of Wikipedia: Knowledge-powered conversational agents.
>
> [2] EMNLP2018 A dataset for document grounded conversations.
>
> [3] EMNLP2018 Towards exploiting background knowledge for building conversation systems.
>
> [4] LLaMA: Open and Efficient Foundation Language Models. 2023.
>
> [5] LoRA: Low-Rank Adaptation of Large Language Models. 2022.

---

### Meta-Review · Area_Chair_DaxT · 2023-09-25

**Recommendation:** 4

**Metareview:**

A new degeneration penalizing objective to improve knowledge internalization (or grounding). Some reviewers expressed their uncertainty about the results, however minor, and they think the paper can be benefitted by incorporating them in the final version.

---

### Decision · Program_Chairs · 2023-10-07

**Decision:**

Accept-Main

**Comment:**

A new degeneration penalizing objective to improve knowledge internalization (or grounding). Some reviewers expressed their uncertainty about the results, however minor, and they think the paper can be benefitted by incorporating them in the final version.